# Mutated *TP53* in Circulating Tumor DNA as a Risk Level Biomarker in Head and Neck Squamous Cell Carcinoma Patients

**DOI:** 10.3390/biom13091418

**Published:** 2023-09-20

**Authors:** Liyona Kampel, Sara Feldstein, Shlomo Tsuriel, Victoria Hannes, Narin N. Carmel Neiderman, Gilad Horowitz, Anton Warshavsky, Leonor Leider-Trejo, Dov Hershkovitz, Nidal Muhanna

**Affiliations:** 1The Head and Neck Cancer Research Laboratory, The Sackler School of Medicine, Tel-Aviv University, 6 Weizman St., Tel-Aviv 6423906, Israel; liyonaka@gmail.com (L.K.); carmeln@gmail.com (N.N.C.N.); 2The Department of Otolaryngology, Head and Neck Surgery and Maxillofacial Surgery, The Sackler School of Medicine, Tel-Aviv University, 6 Weizman St., Tel-Aviv 6423906, Israel; giladhorowitz@gmail.com (G.H.); anton.warshavsky@gmail.com (A.W.); 3The Cancer Research and Pathology Institute, Tel Aviv Sourasky Medical Center, The Sackler School of Medicine, Tel-Aviv University, 6 Weizman St., Tel-Aviv 6423906, Israel; davydova.sara@gmail.com (S.F.); shlomots@tlvmc.gov.il (S.T.); victoriaha@tlvmc.gov.il (V.H.); leonor.trejo@gmail.com (L.L.-T.); dovh@tlvmc.gov.il (D.H.)

**Keywords:** circulating tumor DNA, *TP53*, head and neck squamous cell carcinoma, next-generation sequencing, adjuvant therapy

## Abstract

Circulating tumor DNA (ctDNA) has been suggested as a surrogate biomarker for early detection of cancer recurrence. We aimed to explore the utility of ctDNA as a noninvasive prognostic biomarker in newly diagnosed head and neck squamous cell carcinoma (HNSCC) patients. Seventy HNSCC specimens were analysed for the detection of *TP53* genetic alterations utilizing next-generation sequencing (NGS). *TP53* mutations were revealed in 55 (79%). Upon detection of a significant *TP53* mutation, circulating cell-free DNA was scrutinized for the presence of the tumor-specific mutation. ctDNA was identified at a minimal allele frequency of 0.08% in 21 out of 30 processed plasma samples. Detectable ctDNA correlated with regional spread (N stage ≥ 1, *p* = 0.011) and poorer 5-year progression-free survival (20%, 95% CI 10.9 to 28.9, *p* = 0.034). The high-risk worst pattern of invasion (WPOI grade 4–5) and deep invasion were frequently found in patients whose ctDNA was detected (*p* = 0.087 and *p* = 0.072, respectively). Detecting mutated *TP53* ctDNA was associated with poor progression-free survival and regional metastases, indicating its potential role as a prognostic biomarker. However, ctDNA detectability in early-stage disease and the mechanisms modulating its release into the bloodstream must be further elucidated.

## 1. Introduction

Head and neck squamous cell carcinoma (HNSCC) is the seventh leading cancer worldwide, with approximately 600,000 new cases diagnosed each year [1]. It is comprised of a heterogenous group of tumors that arise from the mucosal lining of the upper aerodigestive tract. In spite of their anatomical and etiological diversity, they are uniformly aggressive, with high rates of progression and relapse despite multimodal therapies [2]. Patients who are diagnosed with advanced-stage HNSCC have an extremely poor prognosis, with five-year survival rates ranging between 25–59% [3,4,5]. The past few decades have witnessed advances in molecular and genetic profiling of cancer. These genomic interrogations have facilitated the emergence of clinically applicable molecular tools, currently under investigation for diagnostic purposes, therapy planning, and disease surveillance in various cancers. 

One promising application is the sampling and analyzing of circulating tumor cells and circulating tumor DNA fragments, that give rise to the concept of liquid biopsy [6]. Circulating tumor DNA (ctDNA), which is comprised of short DNA fragments released into the bloodstream by active secretion or spilled from senescent or apoptotic tumor cells [7,8,9], harbors somatic genetic alternations of the tumor cells that are not present in normal cells [10]. The potential utility of ctDNA as a tumor biomarker has thus far been limited by the low concentration of the mutant cell-free DNA in the bloodstream, sometimes constituting less than 0.01% of circulating DNA [11]. Implementation of new and efficacious sequencing technologies, such as digital droplet polymerase chain reaction (PCR) and next-generation sequencing, has greatly improved ctDNA detection rates [8,12,13]. Several studies have explored the utility of ctDNA for early detection of cancer [14,15], disease monitoring [16,17], evaluation of treatment resistance, and surveillance of residual or recurrent disease [18,19]. ctDNA levels have also been shown to correlate with disease burden and to reliably represent tumor dynamics in breast, colorectal, and pancreatic cancer patients [8,16,17,20]. New evidence has suggested the potential value of ctDNA as a biomarker for the monitoring of treatment or for the detection of residual disease in HNSCC [21,22,23,24]. In the case of virally driven tumors (Epstein–Barr virus-associated nasopharyngeal carcinoma and human papilloma virus [HPV]-associated oropharyngeal SCC), the easily detected viral ctDNA has emerged as an attractive blood-based molecular diagnostic tool [25,26], but its integration into clinical practice is still hindered. It has been only recently reported by Flach et al. [27] in the LIONESS study, which included 17 HPV-negative HNSCC patients, that ctDNA is feasible as a biomarker for detecting disease recurrence before the appearance of any clinical signs. 

In this study, we focused upon somatic alterations of the *TP53* tumor suppressor gene. Previous studies that evaluated the mutational landscape of HNSCC had revealed *TP53* as being a major driver of HNSCC [28,29,30]. The Cancer Genome Atlas reported near-universal *TP53* loss of function mutations in HPV-negative HNSCC, making *TP53* the most frequently mutated gene in HNSCC [28]. The presence of *TP53* mutations was also shown to correlate with treatment response and survival, indicating the potential value of *TP53* analysis for predicting clinical outcomes [29]. 

We speculated that targeted sequencing of *TP53* can detect the specific mutational signatures of most HNSCC tumors. We designed this study to evaluate whether detectable tumor-specific *TP53* alteration in the circulating cell-free DNA in plasma samples obtained immediately before curative-intent surgery could provide an integrative noninvasive molecular biomarker for risk level stratification. Ultimately, we aimed to assess the prognostic value and the potential translational applicability of ctDNA as a surrogate marker for high-risk disease in newly diagnosed resectable HNSCC. 

## 2. Materials and Methods

### 2.1. Study Population 

Approval for this retrospective registry study was granted by the Institutional Review Board at the Tel Aviv Sourasky Medical Center, and informed consent was obtained from all participants. The study period extended from 2014 to 2022. Patients diagnosed with HNSCC were recruited if the samples taken from the oral cavity, pharynx, or larynx had been histologically confirmed as SCC, and if surgical resection planned for curative intent was to take place before the initiation of any oncologic treatment. We excluded patients under 18 years of age, those unable to provide informed consent and patients with distant metastases or other active malignant disease at the time of diagnosis. Cases were also selected from the MIDGAM-Israel National Biobank for Research if plasma samples were available at the time of tumor resection.

### 2.2. Clinical and Histopathological Data and Outcome Measures

Medical records were reviewed and data on demographics, smoking habits, disease characteristics and stage, histopathological findings, and adjuvant therapy were collected and recorded. All tumor specimens underwent a comprehensive histologic assessment by an expert head and neck pathologist. That analysis included documentation of depth of invasion (DOI), perineural or lymphovascular invasion, microscopic measurements of surgical margins, extra-capsular extension, and worst pattern of invasion (WPOI) as described in details elsewhere [31,32]. Briefly, the pattern of invasion Type 1 consisted of tumor invasion in a broad pushing manner, Type 2 represents tumor invading in “finger-like” fashion, and Type 3 demonstrates invasive islands of tumor with >15 cell clusters. Invasive tumor islands smaller than 15 cells per island or strands of tumor cells in a single-cell filling pattern regardless of island size were graded Type 4, and the detection of tumor satellites of any size with a 1 mm or greater distance from tumor/host interface was graded Type 5.

In general, patients were considered for adjuvant radiotherapy if the tumor invasion was deep (≥10 mm), or when metastatic lymph nodes were present. In certain cases, Pathological findings, such as perineural invasion and poor histologic differentiation, in certain cases, were also indications for adjuvant radiotherapy. Unless the patient was medically unfit for chemotherapy, concurrent postoperative radiotherapy and chemotherapy were administered in cases of microscopically involved surgical margins and in cases of extracapsular extension, regardless of the total number of excised metastatic lymph nodes. The latter approach was adopted in accordance with the National Comprehensive Cancer Network guidelines [33]) and based upon the findings of the European Organisation for Research and Treatment of Cancer [34] and Radiation Therapy Oncology Group [35] trials. 

The primary outcome measures were overall survival (OS) considering all-cause mortality and progression-free survival (PFS). Recurrence was defined as the first evidence of recurrent disease or the occurrence of disease-related death. Disease progression was defined when treatment failed, and disease was evident either clinically or radiologically at the first follow-up visit after definitive treatment. 

### 2.3. Tumor DNA Extraction

The hematoxylin and eosin stained FFPE slides of tumor samples were reviewed by a pathologist who identified and marked tumor areas. Serial 8 μm-thick sections of the paraffinized specimen block were placed on slides, and the corresponding marked areas of tumor were micro-dissected with a sterile scalpel. DNA extraction was performed by means of the ReliaPrep FFPE gDNA Miniprep system (Promega, Madison, WI, USA) according to the manufacturer’s protocol. 

### 2.4. TP53 Library Generation and Determining the Fraction of Mutated Copies

Next-generation sequencing was performed with the Ion Torrent Personal Genome Machine (PGM) sequencer platform (Life Technologies, Carlsbad, CA, USA). The DNA purified from the tumor specimens underwent PCR amplification by the use of a panel that covers the *TP53* coding regions (Appendix A). The panel was designed to work in a multiplex reaction according to Biezuner et al. [36], after which the primer sequences were tested with the Thermo Fisher multiple primer analyzer to reduce potential primer dimers. The primer pairs were split into three multiplex mixes, and primers that showed potential dimers or that amplified neighboring areas were assigned to different mixes. 

An amplification-based sequencing method was used to construct libraries for sequencing by means of a two-step PCR protocol. In the first step, the genomic segments were amplified with gene-specific primers. One primer in each pair was coupled to the M13 sequence and the other to the P1 sequence (Appendix A). In the second step, barcodes and adapters were attached to allow their binding to Ion Torrent sphere particles. We aimed for a coverage of at least 1000 reads to enable accurate determination of the mutation fraction in each sample. Sequencing data were initially processed by the Ion Torrent platform-specific pipeline software Torrent Suite (5.2.2). The variant caller results were analyzed by the wAnnovar website [37] in order to identify the significance of each mutation. Mutations were defined by their frequency being above 5% in the sequencing result, and if alteration of the *TP53* protein could be considered as being tumor-related. 

### 2.5. Plasma Sample Collection and ctDNA Detection

Blood samples (8–10 mL) were collected in EDTA collection tubes by standard phlebotomy techniques from recruited patients upon the induction of anesthesia prior to tumor resection and processed within up to two hours for cryopreservation. Blood was centrifuged at 2500 rpm for 10 min at room temperature, and plasma fraction was separated and stored at −80 °C. 

Only patients whose tumor had at least one significant *TP53* mutation, altering protein function with allele frequency above 5%, were included in the analysis of ctDNA. The plasma samples of patients with detectable *TP53* mutation, that had been collected and handled at the time of tumor resection, were thawed and processed for circulating cell-free DNA isolation according to the MagMax Cell Free DNA isolation kit protocol (Life Technologies, Carlsbad, CA, USA). The extracted DNA was quantified by the Qubit DNA high-sensitivity assay kit (Life Technologies, Carlsbad, CA, USA). To detect patient-specific *TP53* mutation, primer pairs were designed to target a short amplicon (70–100 bp) around mutated sequences (primer pairs are shown in Appendix A). Each primer pair was first used to confirm the detection of the mutation in the tumor DNA. Targeted re-sequencing validated all detected mutations. After validation, PCR amplification of the mutated *TP53* position was applied to the DNA extracted from the patient’s plasma, and products were deeply sequenced by the Ion Torrent PGM sequencer. Processing of each sample was carried out in duplicates both on the patient’s cell-free DNA and on wild-type DNA, to determine the level of noise at the specific position of the searched mutation. We aimed for at least 10,000 reads from each sample for accurate determination of mutant fractions in the ctDNA. Mutations were registered when the mutant allele frequency was at least three times higher than noise in the wild-type allele and when each duplicate had a coverage of at least 10 reads of the mutant allele. Mutations were considered negative when they did not meet the above criteria and had a coverage of at least 5000 cumulative reads from the duplicates.

### 2.6. Statistical Analysis

Statistical analyses were performed with GraphPad Prism 8 software (GraphPad Software Inc., La Jolla, CA, USA) and SPSS (IBM SPSS Statistics for Windows, version 27, IBM Corp., Armonk, NY, USA, 2021). Categorical variables were described as frequencies. Odds ratios were used to measure the effect size. The significance and 95% confidence intervals were reported by means of the Fisher exact test. The length of follow-up was calculated with a reverse censoring method. OS and PFS were estimated by Kaplan–Meier survival curves and analyzed by Log Rank tests. All statistical tests were two-sided, and *p* < 0.05 was considered statistically significant. 

## 3. Results

### 3.1. Patient Characteristics

We included 70 tumor specimens of patients diagnosed with HNSCC of the oral cavity, pharynx and larynx treated at the Department of Otolaryngology, Head and Neck Surgery at the Tel Aviv Sourasky Medical Center between January 2014 and April 2022. Formalin-fixed paraffin-embedded (FFPE) slides were retrieved from the institutional pathology archives for DNA extraction, and DNA samples were processed for *TP53* gene sequencing as previously described. The patients included 41 males and 29 females, with a mean age of 65 years (range 35–89 years). None of the patients had positive p16 staining as an indicator of HPV-associated cancer. The majority (61%) were diagnosed with advanced-stage disease (Stages III and IV). Patient and tumor characteristics are depicted in Table 1 and Appendix A. 

### 3.2. TP53 Genetic Alterations in Tumor Specimens

Libraries of the full coding regions of the *TP53* gene were prepared. Overall, we detected one or more significant *TP53* mutations (altering at least one amino acid) in 55 tumor specimens (Appendix A), yielding a *TP53* mutation rate of 79%. The mean coverage obtained for all samples was 1217 aligned reads (range 1000–3853). 

### 3.3. Tumor-Specific TP53 Mutations Detected in Circulating Cell-Free DNA

A plasma sample was collected at the time of tumor resection and available for circulating cell-free DNA analysis in 34 of the 55 patients who had a *TP53* mutation identified in the tumor DNA. All 34 plasma samples were processed for cell-free DNA isolation. Noteworthy, the patients whose plasma samples were analyzed (*n* = 34) and those excluded from ctDNA analysis because a corresponding plasma sample was unavailable (*n* = 21) did not differ in terms of sex, smoking habits, age, tumor site, TNM stage, histopathological features, or postoperative treatment. 

A total of 30 plasma samples had cell-free DNA concentrations sufficient for further processing (>0.1 ng/μL). Mutation-specific primer pairs were designed to target and amplify short amplicons (70–100 bp) around the tumor-specific *TP53* alteration. Amplified PCR products were labelled with barcodes and deeply sequenced by the Ion Torrent PGM (>10,000 reads). The tumor-specific *TP53* genetic alternation was identified in the circulating cell-free DNA of 21 patients (70%), as shown in Table 2. The distribution of mutations along the *TP53* gene that were identified in the tumor specimens and in the matching plasma samples are depicted in Figure 1. 

The cohort of patients included in the circulating cell-free DNA analysis comprised 11 patients with stage I/II disease (37%) and 19 patients with stage III/IV disease (63%). Ten patients were treated for a recurrent or second primary HNSCC. Fourteen patients received adjuvant therapy (seven were treated with radiation and another seven with chemoradiation) in accordance with the National Comprehensive Cancer Network guidelines and the institutional multidisciplinary tumor board discussions. Four patients could not receive adjuvant therapy (three patients underwent salvage laryngectomy and one had previous radiation therapy to the head and neck) and one other patient refused adjuvant therapy. Overall, 8 patients had recurrent disease after curative intent surgery (a median time to recurrence of 22 months, range 9–52 months), and 10 patients showed disease progression despite curative-intent therapy. 

### 3.4. The Prognostic Value of Detectable ctDNA

Detectable ctDNA was significantly associated with regional spread of disease (N stage ≥ 1, *p* = 0.011). Neither the size of the tumor, the pathological T stage, nor the TNM stage correlated with ctDNA detection. We did, however, observe that several pathological adverse features, such as larger DOI and WPOI 4–5, were more frequent among patients whose ctDNA was detected (*p* = 0.072 and *p* = 0.087, respectively). Those associations did not reach a level of significance, probably due to the small number of samples. Interestingly, there were patients with early-stage disease who also had detectable ctDNA but no evidence of regional spread or any pathological adverse features (Figure 2). Patient 9 (Figure 2) underwent partial glossectomy and elective ipsilateral supra-omohyoid neck dissection with radial forearm free flap reconstruction for cT2N0M0 SCC of the tongue. His mutated *TP53* was found in the blood sample drawn at the time of surgical resection. His disease progressed rapidly in spite of the resection having been oncologically complete (negative surgical margins >5 mm) and the absence of any adverse pathological features. There had been no indication for adjuvant therapy postoperatively and the patient died from his cancer shortly after the surgery. Patients 27 and 64 (Figure 2) also had detectable ctDNA in the presence of early-stage disease with no adverse features and they both had disease recurrence. In contrast, patients 33, 52, and 63 (Figure 2) had detectable ctDNA with early-stage disease, but they did not show any signs of disease recurrence or progression. 

The median follow-up time was 16.5 months (range 6–76 months). Kaplan–Meier survival estimates revealed worse PFS when ctDNA was detected (*p* = 0.034), as depicted in Figure 3. A Cox regression model for the estimation of age-adjusted PFS revealed a hazard ratio of 4.36; 95% confidence interval 1.19–15.98 for detectable ctDNA (*p* = 0.026). Interestingly, the presence of *TP53* mutations in tumor samples did not correlate with worse clinical outcomes (Appendix A).

## 4. Discussion 

Despite the great advancement in cancer genomics and sequencing technologies in recent decades, they have not been widely implemented into clinical practice with the aim of improving the clinical outcomes of HNSCC patients. Advanced-stage HNSCC, in particular, portends poor outcomes. New surrogate markers are constantly being sought to guide treatment selection and possibly expand the scope of personalized medicine in the expectation of improving outcomes. ctDNA has been proposed for the monitoring and detection of minimal residual disease in various types of malignant diseases, including breast, colorectal, lung, and pancreatic cancer [16,17,19,38,39]. It has also been shown to reflect the tumor mutational load and to possibly indicate the emergence of resistant clones after therapy [40,41]. Several reports have studied the utility of ctDNA in head and neck cancer [21,22,23,24,42,43,44,45], but evidence remains sparse, with the exception of virus-associated cancers, for which the application of circulating viral DNA for diagnostics and surveillance has demonstrated much progress [25,26]. 

Flatch et al. [27] prospectively analyzed 17 patients with HNSCC of whom 5 had clinical relapse and the detection of ctDNA preceded clinical or CT morphological evidence of disease recurrence. Those authors suggested that the detection of ctDNA as a marker of minimal residual disease following curative-intent surgery holds promise for identifying patients at an increased risk of relapse who may benefit from adjuvant therapy.

We used next-generation sequencing technology to detect *TP53*-mutated ctDNA and to assess its potential value as a biomarker for risk stratification at the time of curative-intent surgical resection. We focused on *TP53* genetic alternations, given that *TP53* is the most frequently mutated gene in HNSCC [28], and one that is also known to carry prognostic significance [29]. In the comprehensive analysis of 70 HNSCC tumors at various stages of the disease, we found at least one significant *TP53* genetic alteration in 79% of the tumor samples, which is consistent with previous studies that reported ~85% somatically mutated *TP53* genes in smoking-related HNSCC [21,28]. The unique mutation detected in each tumor guided the search for the tumor-specific mutation in the circulating cell-free DNA. In the ensuing analysis of 30 matching plasma samples obtained perioperatively, we employed the targeted sequencing approach and found a corresponding *TP53* mutation in the circulating cell-free DNA in 21 cases, representing a ctDNA detection rate of 70%. This approach relies upon prior knowledge of the tumor’s mutational profile to detect the tumor-specific mutation, and it enables a detection rate above the background error rate as opposed to labelling mutations de novo. It also provides high specificity. Comparable rates of ctDNA detection were described by Mes et al. [43] and Harper et al. [45], who reported 67% and 73% detection rates, respectively. However, reported rates are variable, ranging between 40% to 100%, depending upon the genetic profiling methodology and the inclusion of HPV-associated cancer, a factor that bestows favorable detection rates. Perdomo et al. [22] used the targeted approach and reported curative intent ctDNA detection in 42% of their patients, while Wang et al. [21] achieved a 87% detection rate in the plasma and a 96% detection rate when combining plasma and saliva (notably, 21 of their 47 patients had HPV-associated HNSCC). 

Whether or not the ctDNA load represents tumor dynamics is incompletely understood. We had earlier studied the kinetics of ctDNA in an animal model of HPV-associated HNSCC and revealed a correlation between tumor burden and ctDNA levels released into the circulation [46]. We monitored primary tumor and metastatic lymph node volume by computed tomography and found that ctDNA detection could precede evidence of tumor recurrence as depicted on a scan. The sensitivity and specificity of ctDNA detection in that model was 90.2% (95% confidence interval: 76.9–97.3%) and 85.7% (95% confidence interval: 67.3–96.0%), respectively. Correlation between ctDNA loads and disease extent has also been reported in clinical studies on several cancers, with increased amounts of ctDNA having been noted with increased tumor burden [16,38]. Patients with advanced-stage and metastatic cancer were reported to have higher levels of ctDNA [16,20,47]. Mazurek et al. [48] evaluated HPV DNA detection in plasma samples of HNSCC patients and found higher levels of cell-free DNA in patients with clinical N2-N3 disease compared to N0-N1 disease, as well as in patients with stage IV disease compared to stages I-III. In contrast, Wang et al. [21], who reported the largest cohort of HNSCC patients to date, found little difference between the detectability of ctDNA with respect to the stage of disease. When segregated by nodal status, tumor-specific DNA was detectable in the plasma or saliva of 83% (*n* = 59) and 100% (*n* = 34) of their patients with or without nodal metastasis, respectively. Those authors did, however, report higher sensitivity of ctDNA detection for stage III-IV disease (92%, *n* = 37) compared to stage I–II disease (70%, *n* = 10). The association between ctDNA detection and stage of disease may indicate its prognostic utility. Detectable ctDNA in breast and ovarian cancer patients has been found to be a more significant prognostic predictor than commonly used tumor markers [16,49]. In our cohort, the regional spread of disease correlated significantly with the detection of ctDNA (*p* = 0.011). 

We also noted poorer PFS of patients with detectable ctDNA in the blood sampled at the time of surgical resection of the tumor (*p* = 0.034). The high-risk worst pattern of invasion (WPOI 4–5) and depth of invasion were frequently documented in the cases ctDNA was detected (*p* = 0.087 and *p* = 0.072, respectively). Another finding that supports the prognostic value of ctDNA detection at the time of initial surgical resection is that 9 of the 10 patients who relapsed shortly after curative-intent surgery had detectable *TP53*-mutated ctDNA (Figure 2). Specifically, disease progression was noted in two patients with no other clinical or pathological indicators of aggressive disease. Patient 9 with a T2N0 SCC of the tongue (Figure 2) had no pathological adverse feature to justify consideration for adjuvant therapy according to National Comprehensive Cancer Network guidelines. The disease state of patient 57 (Figure 2) with locally advanced T4aN0 laryngeal SCC who underwent total laryngectomy progressed despite adjuvant radiotherapy. Both of those two latter patients might have possibly benefited from more intensified treatment regimens. Contrarily, there were other patients (33, 34, 52, and 63 in Figure 2) who had detectable ctDNA at the time of tumor resection and no evidence of disease progression or relapse during follow-up. 

In order to expand the implementation of ctDNA into clinical practice as a biomarker for risk stratification, it will be necessary to further explore the mechanisms that modulate ctDNA release and clearance from the bloodstream. It is widely accepted that the main sources of ctDNA are cellular breakdown processes, such as apoptosis, necrosis, and autophagy, although active release has also been demonstrated [9,50]. ctDNA levels were found to correspond to hypoxic stress [51], possibly by promoting cell death. Thierry et al. [9] suggested that the total amounts of circulating DNA can represent tumor dynamics and clonal heterogeneity over time and that this association may derive from increased levels of cell death in larger tumor masses. It may also indicate tumor aggressiveness [52], since total circulating DNA concentration was also shown to correlate with OS [53,54]. The association between OS and ctDNA in HNSCC remains unresolved. For example, Harper et al. [45] reported that the presence of ctDNA alterations was associated with decreased OS (hazards rato 3.5, *p* = 0.042), while Perdomo et al. [22] found no such association. We observed a significant correlation between ctDNA detection and worse PFS (Figure 3). 

This study has several limitations. We focused upon the *TP53* gene, which carries high rates of mutations in HNSCC, but this selective approach is expected to detect genetic alterations in only ~85% of cases. Expanding the panel of the tested genes to include PIK3CA, CDKN2A, and NOTCH1 may increase the detection rate to >95%. Having included only a small number of early-stage HNSCC cases might have affected our findings, but, unfortunately, it represents real-life practice since most HNSCC patients are diagnosed with an advanced-stage disease. Although we found that tumor-specific genetic alternations can be detected even when they are low in number, tumor heterogeneity and propagation of a certain sub-clonal population could meaningfully dispute the use of a specific genetic mutation as a diagnostic or prognostic biomarker. To determine if *TP53*-mutated ctDNA can serve as a surrogate marker of disease burden, it would be necessary to determine if treatment results in the clearance of *TP53*-mutated DNA in circulation, which was not included in this study design. Finally, the small sample size likely affected the survival analyses. Further analytical and clinical validation are needed in larger-scale studies.

## 5. Conclusions

ctDNA analysis is already transitioning from the research setting into clinical practice. It has been proposed as a promising prognostic biomarker that defines the subset of patients at high risk of recurrence in colorectal and breast cancer. In the current study, we show that by utilizing highly sensitive genetic sequencing technologies, tumor-specific *TP53* genetic alterations can be identified in the plasma of HNSCC patients upon tumor resection, even at low abundance, and that they are associated with poorer PFS. However, the potential use of *TP53*-mutated ctDNA for risk stratification and guidance of more radical treatment selection must be further evaluated in larger prospective studies prior to its clinical application.

## Figures and Tables

**Figure 1 biomolecules-13-01418-f001:**
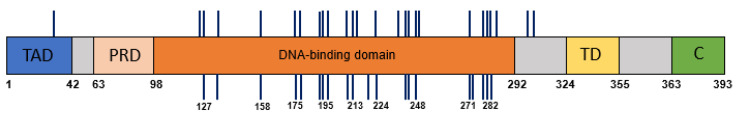
The distribution of mutations across *TP53* gene domains. The mutations identified in HNSCC tumor DNA are represented by vertical lines above the *TP53* gene domains’ illustration. The mutations detected in the plasma samples are depicted below the illustration. Codon numbers are indicated. Abbreviations: TAD, transactivation domain; PRD, proline-rich domain; TD, tetramerization domain; C, C terminus.

**Figure 2 biomolecules-13-01418-f002:**
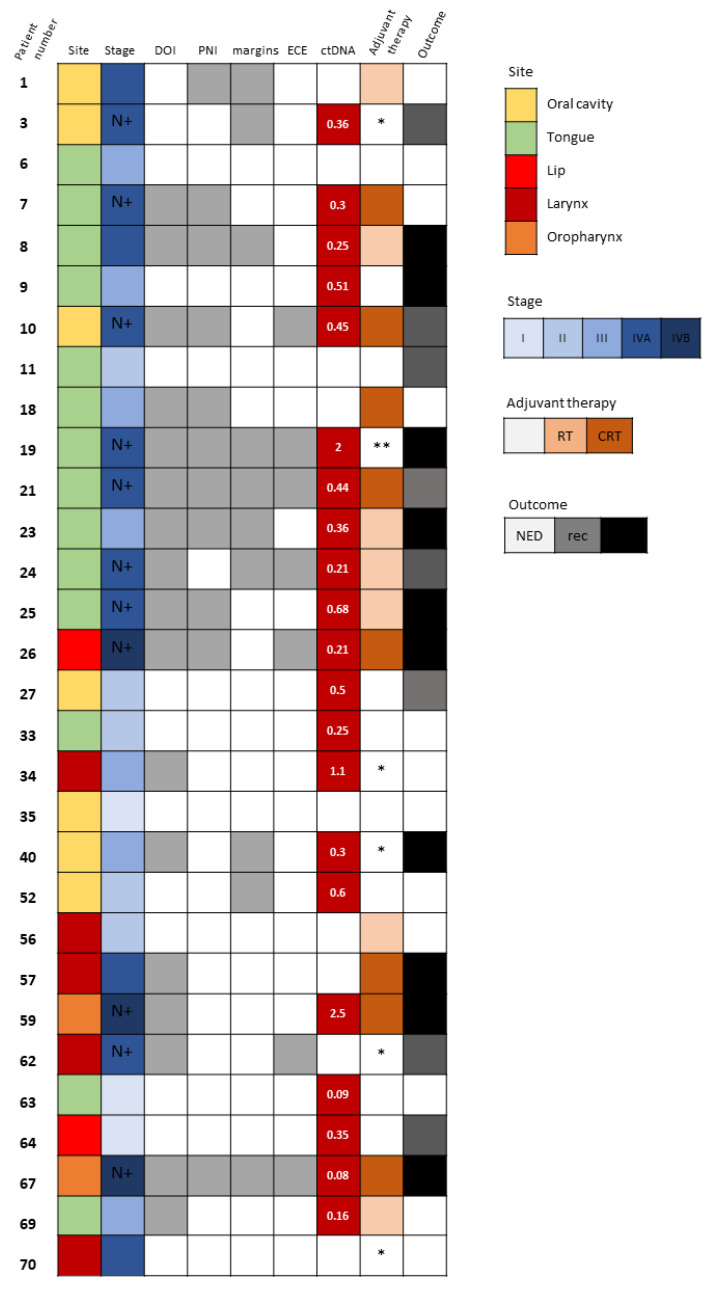
Tumor-specific *TP53* mutation detection in circulating cell-free DNA. Description of clinical and pathological characteristics of 30 HNSCC patients whose matched plasma DNA was scrutinized for the detection of ctDNA based upon the identification of the tumor-specific *TP53* mutation. Site and stage of disease are depicted. N+ denotes patients with N stage ≥ 1. The existence of DOI >10 mm, PNI and close (<5 mm) surgical margins are marked with gray boxes. Levels of detected ctDNA (allele frequency, %) are noted. Adjuvant therapy administered and survival outcomes at last follow-up are shown. In white, NED, no evidence of disease; in gray, recurrence (rec); and in black, disease progression and death. Abbreviations: DOI, depth of invasion; PNI, perineural invasion; ECE, extracapsular extension; ctDNA, circulating tumor DNA; RT, radiotherapy; CRT, chemoradiotherapy. * Not suitable for/refused adjuvant therapy. ** Rapidly progressed before administration of adjuvant therapy.

**Figure 3 biomolecules-13-01418-f003:**
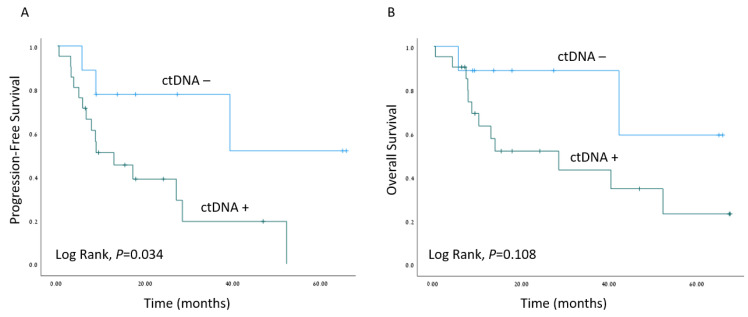
Survival outcomes. Kaplan–Meier survival estimates of progression-free survival (**A**) and overall survival (**B**) of the cohort of HNSCC patients with matched plasma samples (*n* = 30), stratified by ctDNA detection (ctDNA−/ctDNA+).

**Table 1 biomolecules-13-01418-t001:** Patient and disease characteristics (*n* = 70).

Characteristic	*n* (%)
Age (median, range) (y)	65 (35–89)
Male sex	41 (59)
Active/past smoker	39 (56)
Tumor site	
Oral cavity	53 (76)
Oropharynx	5 (7)
Larynx	12 (17)
T stage	
T1	12 (17)
T2	27 (39)
T3	16 (23)
T4	15 (21)
N stage	
N0	39 (56)
N1	6 (9)
N2	17 (24)
N3	8 (11)
Stage (TNM)	
I	10 (14)
II	17 (24)
III	5 (7)
IV	38 (54)

**Table 2 biomolecules-13-01418-t002:** Tumor-specific *TP53* mutations detected in circulating cell-free DNA.

Patient	NA Change	Protein Change	Mutation Type	Circulating cfDNA Concentration (ng/μL)	AF (%)	Noise in WT DNA (%)
1	G811C	E271Q	Missense	0.438	-	0.002
3	G524A	R175H	Missense	0.596	0.36	0.000
6	G856A	E286K	Missense	0.376	-	0.019
7	625_626del	R209Kfs*5	Frameshift del	0.672	0.30	0.031
8			Splicing	0.394	0.25	0.023
9	C742T	R248W	Missense	0.610	0.51	0.000
10	C637T	R213X	Nonsense	0.676	0.45	0.031
11	G473A	R158H	Missense	0.622	-	0.000
18	G524A	R175H	Missense	0.296	-	0.000
19	A583T	I195F	Missense	0.468	2	0.002
21	C380T	S127F	Missense	1.800	0.44	0.001
23	G731T	G244V	Missense	0.416	0.36	0.000
24	C380T	S127F	Missense	0.830	0.21	0.001
25	C844T	R282W	Missense	0.638	0.68	0.019
26	G670T	E224X	Nonsense	1.440	0.21	0.000
27	A138_T140		Non-frameshift	0.438	0.50	0.000
33	A578T	H193L	Missense	0.528	0.25	0.053
34	G730T	G244C	Missense	0.116	1.10	0.000
35	G814A	V272M	Missense	0.443	-	0.002
40	C404T	C135F	Missense	0.282	0.30	0.000
52	A659G	Y220C	Missense	0.342	0.60	0.000
56	C535T	H179Y	Missense	0.345	-	0.000
57	G733A	G245S	Missense	1.200	-	0.000
59	C574T	Q192X	Stopgain	0.602	2.50	0.053
62	C832T	P278S	Missense	1.570	-	0.002
63	C844T	R282W	Nonsense	0.200	0.09	0.019
64	C844T	R282W	Nonsense	0.462	0.35	0.019
67	A536G	H179R	Missense	0.374	0.08	0.000
69	G514T	V127F	Nonsense	0.868	0.16	0.001
70	G743A	R248Q	Missense	0.354	-	0.000

NA, nucleic acid; cfDNA, cell-free DNA; AF, allele frequency; del, deletion; WT, wild type.

## Data Availability

Data will be shared upon reasonable request from the corresponding author.

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
