# Peer review of "Mutated TP53 in Circulating Tumor DNA as a Risk Level Biomarker in Head and Neck Squamous Cell Carcinoma Patients"

_biomolecules, 2023, doi:10.3390/biom13091418_

Round 1

Reviewer 1 Report

In this study, Liyona Kampel analyzed the presence of TP53 mutation in tissue and plasma samples of 70 HPV-negative head and neck squamous cell carcinoma (HNSCC) patients. By next-generation sequencing technology, they analyzed 70 tumor tissue specimens and 35 matched plasma. The main aim of this analysis was to investigate whether circulating tumor  (ctDNA) could be used as a noninvasive prognostic biomarker for risk stratification at the time of curative-intent surgical resection in newly diagnosed HNSCC patients.

TP53 mutations were revealed in 55 (79%) tissue samples and in 21 of the 30 available plasma samples.

In ctDNA, the tumor-specific TP53 genetic alternation was identified at a minimal allele frequency of 0.08%. Detectable ctDNA correlated with regional spread (N stage ≥1, P = .011) and poorer 5-year progression-free survival (20%, 95% CI 10.9 to 28.9, P = .034). The high-risk worst pattern of invasion (WPOI grade 4-5) and deep invasion were frequently found in patients whose ctDNA was detected (P = .087 and P = .072, respectively). Detecting mutated TP53 ctDNA was associated with poor progression-free survival and regional metastases, indicating its potential role as a prognostic biomarker

The subject of this work is certainly of great interest to the Biomolecules journal readership and very timely in general for the potential role of ctDNA in the identification of H&N patients and its clinical management. Data analysis, interpretation, presentation, and discussion are excellent.

To improve the quality of the study, please consider the following comments:

1.                  To further strengthen the study's relevance, it would be valuable to include the detail of mutations found in the tissue and those found in the blood: it would be interesting to see if there is always a correspondence.

2.                  “The tumor-specific TP53 genetic alternation was identified in the circulating cell-free DNA of 21 patients (70%), as shown in Table 2.” I wonder if in the remaining 8 the authors have found different mutations than in the tissue.  If yes, it would be interesting to analyze other pieces of the same biopsy to see if the mutation was not seen just because it was not present in the first piece of the biopsy.

3.                  It would also be very interesting to see if the correlation between mutation presence and PFS is also confirmed in matched tissue or is only in liquid biopsy

4.                  Have the authors tried to combine the two tests to see if the prognostic value increases?

5.                  It is very interesting that the authors found mutations with very low frequencies in plasma. This confirms the ability of some NGS platforms to achieve high resolutions similar to those of dPCR. To confirm this, I would advise the authors to validate some of those mutations with dPCR.

6.                  Since the H&N tumor is characterized by great heterogeneity, I would also suggest analyzing the plasmas of patients in whom no mutation was found. Indeed, it is possible that different mutations may be found in the plasma even if not detected in the tissue (Gangi et al, 2021)

Thanks.

Reviewer 2 Report

1. The results of p53 cfDNA sequencing data showed that only late phase of HNSCC cancer patient’s blood samples which could be assayed with the p53 mutation(s). So it cannot be used to early phase for diagnosis.

2. It better to sequence to key genes for multiple key genes (not only p53) in cfDNA (ex: Cancers 2023, 15(7), 2051) as a index for usage for cancer drug responses perdition or cancer immunotherapy in HNSCC.  

Round 2

Reviewer 1 Report

I thank you the authors for the answers.  

I have no additional comments.

KR

Reviewer 2 Report

This paper only had the assay the p53 ctDNA for HNSCC patients, and the data meaning and the deep of the research is not sutible to pubish in Biomolecules. Only further adding the cancer drug responses perdition or cancer immunotherapy in HNSCC v.s. p53 ctDNA can be publish as the IF>5 (EX: Oncologist. 2022 Jul; 27(7): e604–e605.).  The similar of the study likes this atrilce was published ~IF =3 such as

Acta Oncol . 2020 Jul;59(7):845-850. I had reviewed over then 50 reseach article in over than 20 different jounrals. I highly recommended the authors to resubmit this paper to Genes or Journal of Personalized Medicine.